# The Antioxidant Power of Bergamot Polyphenolic Fraction Gold Potentiates the Effects of L-Citrulline in Athlete Performance and Vasodilation in a Pilot Study

**DOI:** 10.3390/nu17071106

**Published:** 2025-03-21

**Authors:** Rocco Mollace, Roberta Macrì, Maria Serra, Giovanna Ritorto, Sara Ussia, Federica Scarano, Antonio Cardamone, Vincenzo Musolino, Anna Rita Coppoletta, Micaela Gliozzi, Giuseppe Scipione, Cristina Carresi, Kateryna Pozharova, Carolina Muscoli, Francesco Barillà, Maurizio Volterrani, Vincenzo Mollace

**Affiliations:** 1Institute of Research for Food Safety and Health (IRC-FSH), Department of Health Sciences, University “Magna Graecia” of Catanzaro, 88100 Catanzaro, Italy; rocco.mollace@gmail.com (R.M.); maria.serra@studenti.unicz.it (M.S.); giovanna.ritorto@studenti.unicz.it (G.R.); saraussia1598@gmail.com (S.U.); federicascar87@gmail.com (F.S.); tony.c@outlook.it (A.C.); annarita.coppoletta@libero.it (A.R.C.); gliozzi@unicz.it (M.G.); giuseppescipione@icloud.com (G.S.); carresi@unicz.it (C.C.); 0hubabuba0@gmail.com (K.P.); muscoli@unicz.it (C.M.); 2Department of Experimental Medicine, University “Tor Vergata” of Rome, 00133 Rome, Italy; francesco.barilla@uniroma2.it; 3Laboratory of Pharmaceutical Biology, Institute of Research for Food Safety and Health (IRC-FSH), Department of Health Sciences, University “Magna Graecia” of Catanzaro, 88100 Catanzaro, Italy; v.musolino@unicz.it; 4IRCCS San Raffaele Roma, 00163 Rome, Italy; maurizio.volterrani@sanraffaele.it; 5Renato Dulbecco Institute, 88046 Lamezia Terme, Italy

**Keywords:** sport nutrition, supplements, l-citrulline, Bergamot Polyphenolic Fraction Gold (BPFG), nitric oxide, maximal oxygen consumption, running performance, oxidative stress

## Abstract

**Background:** The dietary supplement citrulline might increase nitric oxide levels, leading to vasodilation and improved blood flow, potentially benefiting athletes’ aerobic exercise performance. However, rapid oxidative impairment of the L-arginine/nitric oxide (NO) pathway limits these effects. This is countered by Bergamot Polyphenolic Fraction Gold^®^ (BPFG), a strong natural antioxidant. To investigate L-citrulline + BPFG supplementation’s effects, we performed a randomized, double-blind, placebo-controlled pilot trial on athletic performance and blood flow in trained athletes (cyclists). **Methods:** Random assignment of 90 male athletes resulted in nine different groups: placebo for Group 1, BPFG at 500 and 1000 mg daily for Groups 2 and 3, L-citrulline at 1000 and 2000 mg/daily for Groups 4 and 5, and the combination product of BPFG plus citrulline (N.O. Max) for Groups 6–9. Baseline and 3-month pre- and post-exercise biochemical, reactive vasodilation (RHI), and maximal oxygen consumption measurements were taken for all subjects. **Results:** Three months of the combination of BPFG and L-citrulline (N.O. Max) produced a significant synergistic effect, markedly increasing NO (*p* < 0.001 vs. placebo) release and RHI (*p* < 0.001 vs. placebo). Cardiorespiratory fitness improved significantly with the BPFG and L-citrulline combination, resulting in substantially higher VO_2_ max, VT1, VT2, and peak power and a significantly lower heart rate (*p* < 0.01 vs. placebo). No harmful adverse effects were observed. **Conclusions:** N.O. Max supplementation, providing beneficial effects on the antioxidant state and preserving the vascular endothelium might be a supplementation strategy to improve athletic performance and potentiate results. Given the small sample size, this study serves as a pilot, and further research is needed to validate these findings on a larger scale.

## 1. Introduction

Evaluations of sports nutrition ingredients are needed due to the global sports nutrition market’s size, which reached in USD 45.24 billion in 2023 and is projected to grow by 7.5% from 2024 to 2030 [1,2]. The potential benefits of citrulline for blood flow and exercise performance are causing it to become increasingly popular [3,4,5,6,7]. Because of its potential performance-enhancing effects through improved blood flow, L-citrulline is a common ingredient in 71% of commercial pre-workout supplements. To maximize the performance benefits of L-citrulline, we need to understand its potential effects and then figure out the best time to take it, the right dosage, and if there are any other ingredients that would be good to take with it [3].

Skeletal muscles need more oxygen and energy when they are active during exercise. Enhanced vasodilation and blood flow meet these criteria [8,9,10]. By elevating the nitric oxide (NO) concentration, a potent vasodilator, citrulline may contribute to blood vessel dilation [11,12,13]. Through the citrulline–NO cycle, citrulline is recycled into L-arginine, enabling further NO production. Furthermore, unlike other NO-boosting supplements, L-citrulline does not break down significantly in the body before it reaches the bloodstream, leading to greater increases in plasma L-arginine and improved endurance performance [14,15,16].

In middle-aged individuals, oral L-citrulline supplementation enhanced blood vessel dilation by raising blood L-arginine levels and nitrate excretion [17,18]. Additionally, L-citrulline could affect plasma nitrate levels, potentially lowering the energy demands of exercise by enhancing blood circulation and mitochondrial function [19,20]. Despite this, there is a lack of research on L-citrulline’s effect on NO production and blood vessels in healthy athletes, mainly because NO breaks down quickly due to oxidation.

Supplementation with natural antioxidants, such as green tea, Resveratrol, and micronutrients (calcium, iron, vitamin C, D, and E) has recently been linked to improved athletic performance after exercise, potentially through increased NO production [21,22,23,24,25].

Research has shown that Bergamot Polyphenolic Fraction Gold (BPFG), a bergamot extract that maintains the beneficial compounds of bergamot juice, can improve athletes’ oxygen uptake after exercise [26]. This enhancement is linked to increased nitric oxide release and improved blood vessel dilation exerted by citrus polyphenols, including BPFG [26,27,28]. Indeed, the polyphenols contained in BPF are able to counteract oxidative stress, inflammation, lipid peroxidation, and endothelial dysfunction, inducing an amelioration of NO bioavailability and a reduction in the peroxynitrite concentration [26,27,28].

This suggests that combining BPFG with L-citrulline supplementation could be a practical way to investigate its benefits for athletes.

In this context, the present study aimed to assess the activity of supplementation with BPFG and L-citrulline (N.O. Max) on athletes (cyclists), verifying its antioxidant power and the potential improvement of both sport performance and endothelium-dependent vasodilation.

## 2. Material and Methods

### 2.1. Preparation of BPFG Alone or in Combination with L-Citrulline

Bergamot juice (BJ) was obtained from peeled-off fruits by industrial pressing and squeezing as previously described [29]. The juice was oil fraction-depleted by stripping, clarified by ultra-filtration, and loaded on suitable polystyrene resin columns absorbing polyphenol compounds of MW between 300 and 600 Da (Mitsubishi Chemical, Tokyo, Japan). The polyphenol fractions were eluted by a mild KOH solution [29].

Next, the phytocomplex was neutralized by filtration on cationic resin at acidic pH. Finally, it was vacuum-dried and minced to the desired particle size to obtain BPFG powder. BPFG powder was analyzed by HPLC for flavonoid and other polyphenol content, showing a 47% concentration of naringin, neohesperidine, neoeriocitrine, bruteridine, and melitidine. In addition, toxicological analyses were performed, including for heavy metal, pesticide, phthalate, and synephrine content, which revealed the absence of known toxic compounds. Standard microbiological tests detected no mycotoxins and bacteria (Appendix A).

In conclusion, the BPFG powder (Groups 2 and 3) or L-citrulline (1000 and 2000 mg) will require 500 mg and 1000 aliquots; 2500 mg of maltodextrin were used for treating the placebo group (Group 1). The study participants were given BPFG, L-citrulline, or a placebo in a granulated form that was dissolved in 150 mL of water at 20 °C.

An independent researcher gave the participants the bags containing the test products, unaware of their specific content. L-citrulline had a solubility of 200 mg/mL at 20 °C. The maximum amount that can be dissolved in 150 mL of water is 30 g. Therefore, the L-citrulline powder was fully dissolved.

Dietary supplement procedures were carried out according to the European Community Guidelines.

### 2.2. Study Design

Random assignment of 90 male athletes (cyclists) resulted in nine different groups. Oral supplements were given to nine groups: Group 1 received a placebo, Groups 2 and 3 received 500 and 1000 mg of BPFG daily, Groups 4 and 5 received 1000 and 2000 mg of L-citrulline daily, and Groups 6–9 received a combination of BPFG and citrulline (N.O. Max). For three months, the participants were given daily oral doses of both the placebo and the test products.

Nitric oxide and MDA levels were measured at baseline, 3 months pre-exercise, and 3 months post-exercise in all participants. Baseline and three-month post-supplementation measurements of reactive vasodilation were taken.

Maximal oxygen consumption was assessed via treadmill tests at baseline and following three months of supplementation (Figure 1: Study design).

After an overnight fast, all clinical data was collected, including biochemical parameters and ultrasound examination. Echocardiograms, Endopat evaluations, peripheral blood collection for biochemical analysis, and physical exercise tests were conducted on members of each group at the start and after three months of supplementation, as previously described. Table 1 displays the mean ± SD for age, height, body mass, and body mass index of the participants. Additionally, ECG measurements of athletes reveal that left ventricular structure and function were within normal limits. Left ventricular muscle mass, interventricular septum diameter during diastole, and left ventricular posterior wall thickness during diastole were determined for all subjects through M-mode and 2D Doppler–ECG imaging using an ARTIDA ultrasound system (Toshiba, Tokyo, Japan) with standard transducers. LVMI was determined by adjusting for body surface area (Table 1 and Table 2).

The study excluded participants with hemodynamic dysfunction, inflammatory diseases in the past three months, and cigarette smoking, as these factors could affect vascular parameters. No subjects were taking citrus products, supplements, or medications that might affect their cardiovascular system. The participants were advised to avoid exercise for 24 h prior to the ultrasound measurements.

No caffeine, antioxidants, or alcohol were permitted for 48 h before the experiment. Three weeks prior to the study, all participants were put on a mixed, isocaloric diet (2800 ± 800 kcal/day) consisting of carbohydrates in the amount of 365.3 ± 152.6 g/day, proteins: 130 ± 45.5 g/day, and fats: 109 ± 45 g/day (monounsaturated fats: 35 ± 15 g/day and polyunsaturated fats: 8.2 ± 4 g/day).

Compliance and adverse effects were monitored biweekly. Participants followed an isocaloric diet supplemented with a placebo, BPFG, citrulline, or a combination. To make sure that the subjects consumed a diet, they were asked to sign a statement of compliance with the diet. Physiologists and the coach of athletes regularly reminded them about the need to comply with the dietary regime.

The athletes returned unused granulated sachets; investigators or their designees count and record these at the end of each visit and after enrollment. An automated calculation of compliance has been performed, using the number of sachets taken and the expected number. Unless the athlete returned the remaining granulated sachets, the investigator estimated how many there were. Athletes with compliance below 80% are questioned to determine the cause. In particular, the possibility of adverse events has been evaluated. No harmful adverse effects were observed.

The study was approved by the Ethics Committee of the University of Catanzaro and conformed to the standards set by the Declaration of Helsinki (Calabrian Region—Protocol Registry n. 387, 24 September 2022).

### 2.3. Endothelial Function Assessment

Endothelial function was assessed using the EndoPAT 2000 technique, which measures PAT using the reactive hyperemia index (RHI, arbitrary units) [30]. Briefly, after 20 min of rest in a chair inclined at an angle of about 45° at room temperature, a blood pressure cuff was placed on the non-dominant upper arm (study arm), while the other arm served as the control. The hands were placed on armchair supports with the palm side down, so the fingers hung freely. Next, they positioned the EndoPAT probes on the tips of both index fingers. Without touching any other finger or object, the probes were electronically inflated. A personal computer continuously captured the PAT signal during the entire test. Measurements of baseline pulse amplitude at each fingertip were taken over a five-minute period. A five-minute baseline recording was taken on both arms. Blood flow was halted in the experimental arm by inflating the cuff to 200 mmHg or 60 mmHg above the systolic blood pressure, whichever was greater. The cuff pressure was swiftly reduced after a five-minute occlusion, with post-occlusion recordings continuing for another five minutes in both the experimental and control groups. The RH–PAT or PAT ratio was calculated by dividing the average pulse amplitude after deflation in the experimental finger by the hyperemia level in that finger. Endothelial function was measured using the RHI, and arterial stiffness was assessed by the EndoPAT 2000 (Itamar Medical Ltd., Caesarea Ind. Park, Israel) through PAIx from radial pulse wave analysis. Calculated as a percentage, PAIx represents the ratio of late systolic pressure to early systolic pressure.

### 2.4. Blood Collection and Biochemical Analyses

At the beginning of the study (pre-intervention) and at the end of each treatment period (after 3 months), all subjects reported to the laboratory and had venous blood drawn for the determination of total NO, nitrite/nitrate ratio, and malondialdehyde (MDA) concentrations. The blood was allowed to clot at room temperature and then was centrifuged to obtain the serum, which was aliquoted and frozen at −80 °C until the assays. According to the manufacturer’s protocol, the oxidative stress index was assessed by plasma lipid peroxidation product malondialdehyde (MDA) through a lipid peroxidase assay kit (Sigma-Aldrich, Saint Louis, MO, USA). Briefly, the lipid peroxidation is determined through the reaction of MDA with thiobarbituric acid (TBA) to obtain a colorimetric (532 nm)/fluorometric (λex = 532/λem = 553 nm) product, proportional to the MDA concentration in the sample [31].

### 2.5. NO and Nitrite/Nitrate Ratio Assay

Nitric oxide concentrations are measured in serum samples with the Total Nitric Oxide and Nitrate/Nitrite Assay by converting nitrate to nitrite through the enzymatic action of nitrate reductase. Following the reaction, nitrite is measured colorimetrically as an azo dye, a product of the Griess reaction. The Griess reaction involves a two-step diazotization process in which an acidified NO_2_^−^ generates a nitrosating agent, which then reacts with sulfanilic acid to yield the diazonium ion. Coupling this ion with *N*-(1-naphthyl) ethylenediamine produces the chromophoric azo derivative, absorbing light in the 540–570 nm range [32].

### 2.6. Body Mass Index Assessment

BMI calculations were used to assess the enrolled patients for overweight, obesity, and weight changes. Standard procedures were followed to measure the body weight and height of all participants in a fasting state, standing upright, and without shoes. The Radwag C315.100/200 OW digital personal scale from Radom, Poland, offered measurements accurate to 0.1 kg/0.1 cm for body weight and height. The body mass index (BMI) is a useful statistical tool for estimating body fat based on weight and height and is suitable for both males and females of any age. Specifically, BMI is calculated by dividing weight in kilograms by height squared in meters. A normal range of 18.50–24.99 kg/m^2^ was determined. BMI values between 25 and 29.9 kg/m^2^ are considered overweight, while values exceeding 30 kg/m^2^ indicate obesity [33].

### 2.7. Exercise Test

A standard treadmill exercise test was conducted before and after each three-month treatment protocol (supplementation or placebo) to assess if BPFG enhances the performance of L-citrulline in athletes [34]. The test started with a 3 min warm-up, after which intensity was augmented by 40 W every 3 min until the participants reached their highest exertion level. From six minutes prior to the exercise and throughout each stage of the exercise load, continuous measurements of pulmonary ventilation, oxygen uptake (VO_2_), ventilatory thresholds (VT1 and VT2), and carbon dioxide output (CO_2_) were collected using the Oxycon Apparatus (Erich Jaeger GmbH, Hoechberg, Germany). Continuous heart rate (HR) monitoring (PE-3000 Sport-Tester, Polar Inc., Kempele, Finland) and duplicate blood pressure (SBP/DBP) measurements with a sphygmomanometer occurred before and after exercise.

### 2.8. Statistical Analysis

Each result is shown as the average value with the standard deviation represented as a range. We compared the baseline and post-exercise variables before and after the intervention (placebo or BPFG, L-citrulline, and BPFG + l-citrulline (N.O. Max)). We determined the change in baseline NO levels and the increase in maximal oxygen uptake (DV_O_2_max) before and after the intervention. An analysis of variance (ANOVA) was used to analyze the data. The normally distributed data were analyzed by a one-way ANOVA followed by Tukey’s test, while the data without normal distribution were analyzed using a Kruskal–Wallis analysis of variance and Dunn’s tests. For all calculations, the significance level (alpha) and power are set at the values of 0.05 and 80.0%, respectively. The statistical power was limited due to the small sample size in this pilot study.

## 3. Results

### 3.1. Protective Effect of BPFG and L-Citrulline Combination on NO Bioavailability and Endothelial Dysfunction

The supplementation of athletes with BPFG (500 and 1000 mg/once day for 3 months; Groups 2 and 3) compared to placebo (Group 1) produced a significant increase of total NO and nitrite/nitrate ratio, an effect that was associated with enhanced RHI (reactive vasodilatation), as detected via Endopat analysis. This effect occurred both at rest and after exercise (Figure 2 and Figure 3). Groups 4 and 5, receiving 1000 and 2000 mg/daily L-citrulline supplementation for 3 months, both showed comparable outcomes. Athletes supplemented with L-citrulline showed significant increases in both NO release and reactive hyperemia index (RHI) at both pre- and post-exercise (Figure 2 and Figure 3).

Interestingly, the supplementation with BPFG plus L-citrulline (1000 and 2000 mg/daily, Group 6 to 9, respectively) resulted in a highly significant enhancement of NO release and a more pronounced vasodilatation index compared to placebo (*p* < 0.001) (Figure 2 and Figure 3). The effect was observed in measurements taken both before and after exercise.

### 3.2. The Beneficial Effects of N.O. Max Supplementation on Exercise Performance for Athletes

Exercise performance was significantly enhanced when BPFG and L-citrulline (N.O. Max^®^) were combined, as shown by treadmill tests. L-citrulline, when used independently, did not demonstrate considerable advantages for athletes (as shown in Table 3). Combining BPFG (500 mg/daily and 1000 mg/daily) with L-citrulline, however, resulted in highly significant improvements in VO_2_max, VT1, VT2, peak power, and HR for athletes when compared to using BPFG or L-citrulline separately (Table 3).

### 3.3. N.O. Max Supplementation Decreases Oxidative Stress in Athletes

Similar findings were observed for MDA levels, a reliable indicator of oxidative processes, both during rest and post-exercise (Figure 4). Notably, increased MDA levels were observed mainly in athletes following exercise. BPFG (Groups 2 and 3) reduced very significantly the MDA levels, and a minor reduction was observed for L-citrulline (Groups 4 and 5). The combination of BPFG and L-citrulline (Groups 6 to 9) showed a highly significant reduction in MDA levels in a dose-dependent manner, indicating BPFG’s ability to mitigate oxidative processes, which inhibit NO activity in athletes, thereby enhancing L-citrulline’s positive effects on post-exercise recovery and performance (Figure 4).

## 4. Discussion

This clinical study aimed to determine whether combining BPFG and L-citrulline supplements could have a positive combined effect on athletes’ (cyclists) blood vessel health, muscle function, and nitric oxide levels [2]. The study revealed that taking BPFG and L-citrulline together for three months resulted in higher serum NO levels, both before and after exercise, compared to taking each supplement alone. The heightened asymmetric dimethylarginine (ADMA) levels, a marker of NO production, in athletes treated with BPFG plus L-citrulline confirmed this effect. Furthermore, both effects were associated with increased reactive vasodilation, as evidenced by the RHI readings both before and after exercise. While L-citrulline increased NO production, it failed to meaningfully enhance athletes’ performance after exercise. The treadmill test showed that L-citrulline had no significant impact on VO_2_ max or peak response. VT1, also known as the aerobic threshold, marks the point where the body begins to rely on both aerobic and anaerobic energy production instead of primarily aerobic energy [35]. The second ventilatory threshold, also known as the lactate threshold, is referred to as VT2 [36]. The maximum amount of oxygen that the body can utilize in a minute per kilogram of body weight is referred to as VO_2_ max [37]. The combination of BPFG and L-citrulline significantly improved aerobic capacity, oxygen uptake, and lactate threshold by increasing VO_2_ max, VT1, and VT2; these results were followed by a significant Peak Power output increase, with a better HR response. Our research findings align with previous studies indicating that BPFG supplementation enhances NO bioavailability by safeguarding NO from oxidative damage. Thus, we proposed that BPFG could decrease the strain on the cardiovascular system and enhance athletic performance by altering the NO levels in the blood of the skin and skeletal muscles. Several pathways have been proposed for BPFG’s vasoprotective effects [38,39], including enhanced fat oxidation in the liver and muscles [40] and the maintenance of vasodilation through stimulating NO production [41,42]. Our data reveal that BPFG and L-citrulline together decrease MDA levels, suggesting BPFG’s antioxidant effects enhance L-citrulline’s impact. However, their combined effect on exercise performance needs further investigation. The enhanced NO production and the promoted degradation of its radical, resulting from 3 months of BPFG supplementation, may significantly contribute to the observed differences compared to L-citrulline alone. Indeed, improved endothelial function results from its role in maintaining vascular function and structure in response to exercise [43,44,45]. Previous data on oxidative stress and endothelial function in athletes and non-athletes completely supports this [46]. Evidence suggests that antioxidant supplements significantly boost exercise performance in athletes [21,22,47,48,49,50]. Specifically, the data show that *N*-acetylcysteine (NAC) supplementation significantly improves athletes’ endurance [51,52] Increased antioxidant availability and better K^+^ control, possibly through the Na+/K+ pump, could be behind the performance improvement [53]. We also observed that a stronger peak response during treadmill tests in athletes using BPFG plus L-citrulline correlated with higher VO_2_ max and improved HR response. Conversely, our findings align with evidence indicating that NO-donor supplementation, while increasing NO levels is ineffective at improving exercise performance unless paired with agents that mitigate oxidative stress [54].

A growing body of evidence indicates that high-intensity exercise leads to exercise-induced oxidative stress [55,56]. Intense and prolonged exercise, specifically, highlights the vital role that free radicals play in many physiological processes and functions [57]. However, the combination of these with NO creates dangerous peroxynitrite and other free radicals that can harm blood vessels and impair muscle performance [58,59]. Alone, L-citrulline supplementation does not significantly improve athletes’ muscle performance under these conditions. However, combining antioxidant supplements (BPFG) with citrulline supplementation can restore the mechanisms that promote a healthy NO–endothelium response, thereby improving muscle function during exercise. Conversely, BPFG seems to offset the failure of the body’s natural antioxidant defenses against persistent oxidative stress. According to recent studies, polyphenols play a key role in the regulation of specific cellular mechanisms by promoting endothelial NO synthesis, which is an intracellular and extracellular messenger, resulting in enhancing blood flow and vasodilatation [49,60]. Therefore, in relation to these remarkable vascular effects, the most plausible mechanism is probably the reduction of ROS generation or the enhancement of ROS detoxification capacity via the antioxidant system. The bioavailability of NO could be improved by reducing exposure to ROS, which decreases peroxynitrite production due to the reaction between superoxide and NO [60].

Therefore, polyphenol supplementation in sports disciplines could potentially improve athletic performance by ameliorating endothelial function and vasodilation, which is a result of increasing hemodynamic function [49,60]. 

The limitations of the study originate from the general limitations of a pilot study: a small sample size and a statistical lack of power. Long-term follow-up was lacking as well. Indeed, this study serves as a pioneer study, examining the effects of N.O. Max supplementation, particularly in view of athletic performance and NO bioavailability regulation [61]. Further research is needed to validate these findings on a larger scale, particularly regarding the potential dose–response relationship and long-term effects.

## 5. Conclusions

In conclusion, this pilot clinical trial showed, for the first time, that athlete (cyclist) supplementation with BPFG and L-citrulline (N.O. Max) enhances NO release, leading to antioxidant effects and improved endothelium-dependent vasodilation. BPFG and L-citrulline supplementation improves skeletal muscle blood flow and oxygen uptake in athletes, thus improving sports performance. Upon confirmation from extensive multicenter studies with extended follow-up periods, N.O. Max might represent an ideal supplement for athletes by optimizing the nitric oxide levels to support the flow of blood and oxygen to the endothelial and skeletal muscles, thus supporting a significant synergistic effect on vasodilation and sports performance.

## Figures and Tables

**Figure 1 nutrients-17-01106-f001:**
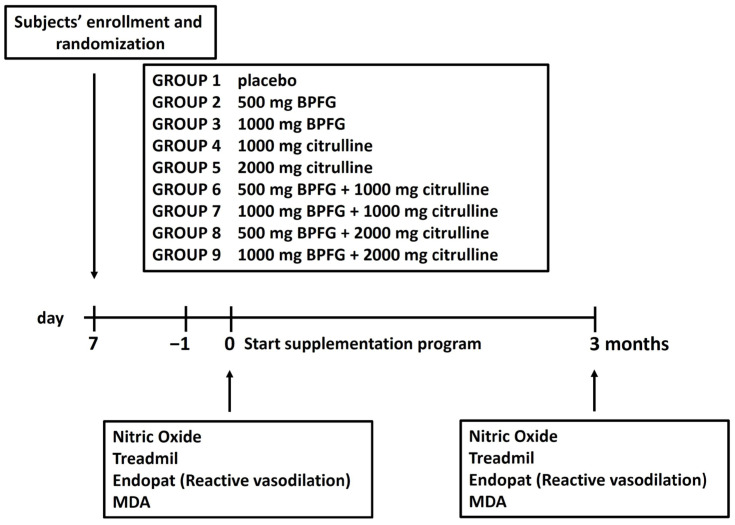
Study design.

**Figure 2 nutrients-17-01106-f002:**
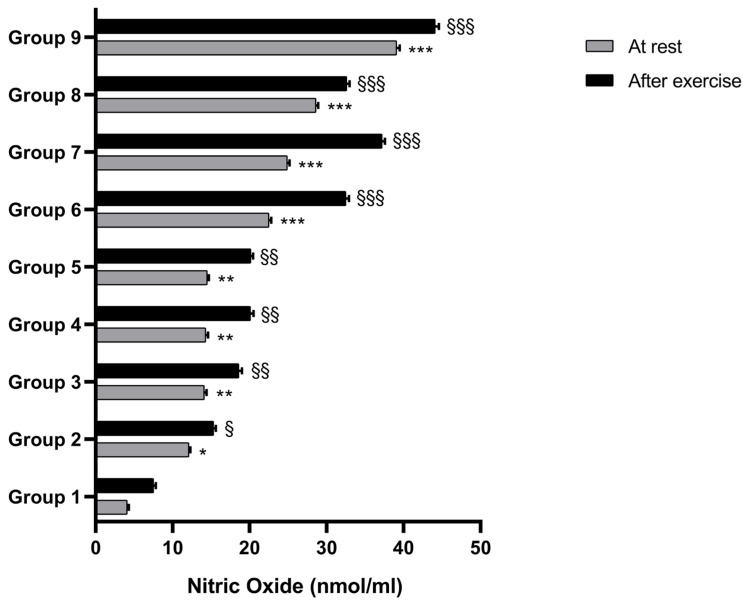
The effects of BPFG, citrulline, and BPFV + citrulline (N.O. MAX) in nitric oxide release at rest and after exercise. * *p* < 0.05 group 2 vs. group 1 (placebo rest); ** *p* < 0.01 group 2, 3, 4 and 5 vs. group 1 (placebo rest), respectively; *** *p* < 0.001 group 6, 7, 8 and 9 vs. group 1 (placebo rest), respectively; § *p* < 0.05 group 2 vs. placebo group 1 (exercise). §§ *p* < 0.01 group 3, 4, 5 vs. placebo group 1 (exercise), respectively. §§§ *p* < 0.001 group 6, 7, 8 and 9 vs. vs. placebo group 1 (exercise).

**Figure 3 nutrients-17-01106-f003:**
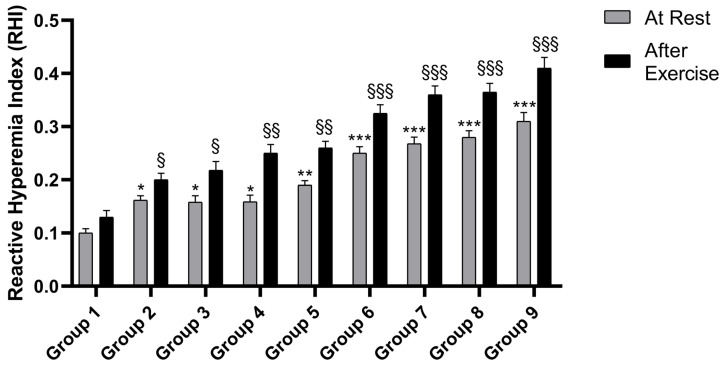
The effect of BPFG, L-citrulline, and BPFG + L-citrulline (N.O. Max) on RHI at rest and after exercise. * *p* < 0.05 group 2, 3, 4 vs. group 1 (placebo rest) respectively; ** *p* < 0.01 group 5 vs. group 1 (placebo rest); *** *p* < 0.001 group 6, 7, 8 and 9 vs. group 1 (placebo rest), respectively; § *p* < 0.05 group 2, 3 vs. placebo group 1 (exercise), respectively; §§ *p* < 0.01 group 4, 5 vs. placebo group 1 (exercise), respectively. §§§ *p* < 0.001 group 6, 7, 8 and 9 vs. vs. placebo group 1 (exercise).

**Figure 4 nutrients-17-01106-f004:**
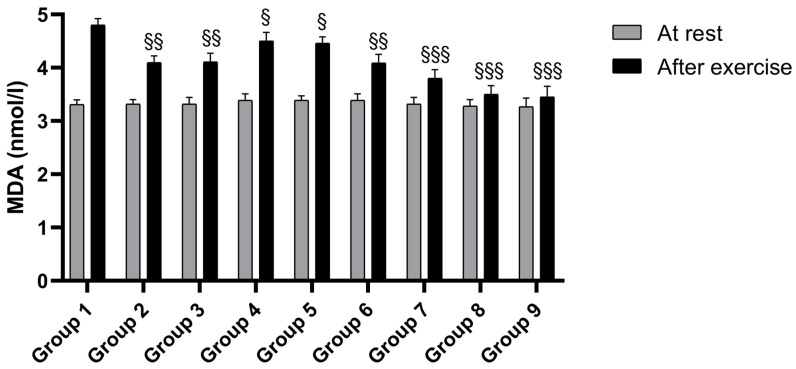
The effect of BPFG, L-citrulline, and BPFG + L-citrulline (N.O. Max) on MDA serum levels at rest and after exercise. §: *p* < 0.05 vs. group 1 (placebo exercise), respectively; §§: *p* < 0.01 vs. group 1 (placebo exercise); §§§ *p* < 0.001 vs. group 1 (placebo exercise).

**Table 1 nutrients-17-01106-t001:** Anthropometric variables in the study population (n = 90 athletes; values are expressed as mean ± SD).

Groups	Age (years)	Body Weight (kg)	Body Height (cm)	BMI (kg/m^2^)
Group 1 (n = 10)	32 ± 4	70.5 ± 1.9	180.2 ± 2.3	23 ± 1.6
Group 2 (n = 10)	32.5 ± 2.8	71.2 ± 2	180 ± 1.6	22.8 ± 2.1
Group 3 (n = 10)	31.9 ± 3	72.1 ± 3	181.2 ± 3.1	24 ± 1.8
Group 4 (n = 10)	32 ± 3.6	71.8 ± 2.4	180 ± 1.9	23 ± 1.4
Group 5 (n = 10)	33 ± 4.1	71.3 ± 2.3	181.1 ± 2.5	23.7 ± 1.7
Group 6 (n = 10)	32 ± 3.5	72 ± 2.8	180.5 ± 2.4	22.8 ± 1.4
Group 7 (n = 10)	31.9 ± 3.2	70.7 ± 2	180.1 ± 2.2	22.9 ± 1.3
Group 8 (n = 10)	31.9 ± 3.1	72.1 ± 2.8	179.5 ± 1.7	23.1 ± 1.5
Group 9 (n = 10)	32 ± 3.4	71 ± 2.5	179.3 ± 1.5	23.5 ± 1.6

BMI, body mass index.

**Table 2 nutrients-17-01106-t002:** Echocardiographic variables in the study population (n = 90 athletes; values are expressed as mean ± SD).

Groups	LVM (g)	LVMI (g/m^2^)	IVSDd (mm)	LVPWTd (mm)	LVEF (%)	SV (mL)
Group 1 (n = 10)	222 ± 12.2	124 ± 11.7	11 ± 1.1	9 ± 1	62 ± 2.1	80 ± 3.9
Group 2 (n = 10)	221.5 ± 11.9	125 ± 12.5	10.5 ± 1.2	8.8 ± 0.2	62.5 ± 2.2	80.2 ± 4.2
Group 3 (n = 10)	223 ± 12.6	123.6 ± 11.4	11.5 ± 1.3	9 ± 1	62.8 ± 2.2	81 ± 4.5
Group 4 (n = 10)	222 ± 12.1	123.7 ± 11.8	12 ± 1.4	8.9 ± 0.6	62 ± 2.1	79.2 ± 3.7
Group 5 (n = 10)	223.1 ± 12.5	124 ± 12	11.1 ± 1.8	10.3 ± 2	61.5 ± 1.9	80 ± 4
Group 6 (n = 10)	221.7 ± 12	124.5 ± 12.1	10.8 ± 1.3	9.1 ± 1.2	61.2 ± 1.8	79.6 ± 3.8
Group 7 (n = 10)	221.8 ± 12	124.3 ± 12	12 ± 1	9.4 ± 1.4	63 ± 2.4	80.7 ± 4.3
Group 8 (n = 10)	222.2 ± 12.3	125.3 ± 12.5	11 ± 1	10.3 ± 1.6	61.8 ± 2.1	80.2 ± 4.1
Group 9 (n = 10)	222.7 ± 12.4	125.3 ± 13	10.9 ± 1.9	9.8 ± 1.8	62.2 ± 2.2	81.1 ± 4.5

LVM, left ventricular mass; LVMI, left ventricular mass index; IVSDd, intraventricular septum diameter during diastole; LVPWTd, left ventricular posterior wall thickness during diastole; LVEF, left ventricular ejection fraction; SV, stroke. volume.

**Table 3 nutrients-17-01106-t003:** The table shows the effect of BPFG (Group 2 and 3, 500 and 1000 mg, respectively), L citrulline (Group 4 and 5, 1000 and 2000 mg, respectively), and their combination product (NO Max, Groups 6–9) on VO^2^ Max (mg/Kg/min), VT1 and VT2 (mL/Kg/min), peak power (Watt) and HR Max (bpm), after treadmill before and at the end of treatment compared to placebo (Group 1). ^§^: *p* < 0.05 vs. group 1 (placebo exercise), respectively; ^§§^: *p* < 0.01 vs. group 1 (placebo exercise); ^§§§^ *p* < 0.001 vs. group 1 (placebo exercise).

VO_2_ Max (mL/kg/min)									
	Group 1	Group 2	Group 3	Group 4	Group 5	Group 6	Group 7	Group 8	Group 9
**Basal**	68 ± 2.8	68 ± 2.4	66 ± 2.5	65 ± 2.3	67 ± 2.6	68 ± 3.3	65 ± 2.5	66 ± 2.7	68 ± 3.1
**Post-** **treatment**	67 ± 2.2	74 ± 3.4 ^§^	76 ± 2.5 ^§^	74 ± 2.6 ^§^	76 ± 2.8 ^§^	81 ± 2.1 ^§§^	89 ± 3.1 ^§§§^	84 ± 2.9 ^§§^	96 ± 3.2 ^§§^
**Peak Power (watt)**									
	**Group 1**	**Group 2**	**Group 3**	**Group 4**	**Group 5**	**Group 6**	**Group 7**	**Group 8**	**Group 9**
**Basal**	415 ± 6.0	420 ± 5.2	416 ± 6.2	422 ± 10	412 ± 6.5	419 ± 5.4	421 ± 5.2	415 ± 5.2	425 ± 6.3
**Post-** **treatment**	425 ± 6.3	445 ± 7.4 ^§^	425 ± 14 ^§^	432 ± 4.6 ^§^	438 ± 13 ^§^	472 ± 15 ^§§^	486 ± 4.8 ^§§§^	474 ± 6.5 ^§§^	492 ± 5.9 ^§§§^
**HR Max (beats/min)**									
	**Group 1**	**Group 2**	**Group 3**	**Group 4**	**Group 5**	**Group 6**	**Group 7**	**Group 8**	**Group 9**
**Basal**	160 ± 2.5	162 ± 3.3	158 ± 3.0	157 ± 3.1	160 ± 3.2	155 ± 2.2	160 ± 3.2	160 ± 3.2	160 ± 3.3
**Post-** **treatment**	158 + 2.2	156 ± 2.6 ^§^	155 ± 3.1 ^§^	154 ± 2.5 ^§^	152 ± 3.1 ^§^	150 ± 2.5 ^§§^	148 ± 3.0 ^§§^	148 ± 3.0 ^§§^	144 ± 3.2 ^§§^
**VT1 (mL/Kg/min)**									
	**Group 1**	**Group 2**	**Group 3**	**Group 4**	**Group 5**	**Group 6**	**Group 7**	**Group 8**	**Group 9**
**Basal**	16.7 ± 1.8	17.1 ± 2.1	16.2 ± 1.6	16.2 ± 2	16.1 ± 2.3	17.2 ± 1.8	16.5 ± 1.9	15.9 ± 2.2	16.8 ± 1.6
**Post-** **treatment**	16.4 ± 1.9	20.2 ± 1.8 ^§^	20.9 ± 2.1 ^§^	21.3 ± 2.2 ^§^	22.2 ± 2.1 ^§^	23.1 ± 2.4 ^§§^	23.9 ± 2.4 ^§§^	24.4 ± 1.9 ^§§^	25.8 ± 2.7 ^§§^
**VT2 (mL/Kg/min)**									
	**Group 1**	**Group 2**	**Group 3**	**Group 4**	**Group 5**	**Group 6**	**Group 7**	**Group 8**	**Group 9**
**Basal**	27.2 ± 2.5	26.9 ± 2.1	27.3 ± 2.6	27.4 ± 2.7	26.8 ± 2.4	27.3 ± 2.7	27.2 ± 2.5	2.75 ± 2.4	27.0 ± 2.6
**Post-** **treatment**	27.4 ± 2.8	30.8 ± 2.8 ^§^	31.6 ± 2.7 ^§^	31.8 ± 2.4 ^§^	32.5 ± 2.7 ^§^	32.4 ± 2.6 ^§§^	33.5 ± 2.9 ^§§^	34.5 ± 2.8 ^§§^	36.7 ± 2.2 ^§§§^

VO_2_: volume of oxygen; VT1: ventilatory thresholds 1; VT2: ventilatory thresholds 2; HR: heart rate.

## Data Availability

The data presented in this study are available upon request from the corresponding author. The data are not publicly available for privacy and ethical reasons.

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
