# Peer review of "The Antioxidant Power of Bergamot Polyphenolic Fraction Gold Potentiates the Effects of L-Citrulline in Athlete Performance and Vasodilation in a Pilot Study"

_nutrients, 2025, doi:10.3390/nu17071106_

Round 1

Reviewer 1 Report

Comments and Suggestions for Authors

Title of the Manuscript: Antioxidant properties and enhanced vasodilation contribute to the potentiation of L-citrulline effects in athletes mediated by combination with Bergamot Polyphenolic Fraction Gold.

Journal: Nutrients (Q1 Journal)

Overall Assessment: This manuscript presents a well-structured and scientifically relevant study on the combined effects of L-citrulline and Bergamot Polyphenolic Fraction Gold (BPFG) on vasodilation, nitric oxide (NO) release, and athletic performance. The study is based on a robust randomized, double-blind, placebo-controlled design, ensuring the reliability of the findings. The results contribute valuable insights into sports nutrition, particularly regarding the synergistic effects of BPFG and L-citrulline. However, some aspects of the manuscript require improvement, including methodological clarifications, statistical interpretations, and discussion coherence.

  1. Abstract

Strengths:

  • Clearly outlines the study's background, objectives, methodology, key findings, and conclusions.
  • Provides a succinct summary of major results, particularly emphasizing the synergistic effects of BPFG and L-citrulline.

Areas for Improvement:

  • The abstract would benefit from including explicit p-values or effect sizes to strengthen the statistical support of the claims.
  • The concluding statement should explicitly state the practical applications of the findings in sports nutrition and performance enhancement.
  1. Introduction

Strengths:

  • Well-contextualized within the broader field of sports nutrition.
  • Clearly identifies the gap in knowledge regarding the oxidation-related limitations of L-citrulline and the potential benefits of BPFG.
  • Provides a comprehensive literature review supporting the rationale for the study.

Areas for Improvement:

  • The introduction should include more discussion on the biochemical mechanisms underlying BPFG’s potential protective effects against oxidative degradation of NO.
  • Some claims (e.g., those regarding the prevalence of L-citrulline in commercial supplements) would benefit from more recent references.
  1. Methods

Strengths:

  • The study design is well-explained, including details on randomization, blinding, and placebo control.
  • Clear description of the biochemical and physiological assessments, including NO bioavailability, endothelial function, and exercise performance measures.

Areas for Improvement:

  • The sample size determination should be explicitly stated with a justification based on power analysis.
  • The methodology for statistical analysis (e.g., justification for ANOVA over other statistical models) requires more detail.
  • Details on compliance monitoring for supplementation adherence should be included.
  1. Results

Strengths:

  • Data are clearly presented with appropriate figures and tables.
  • The results are systematically reported, following the logical sequence of primary and secondary outcomes.
  • The NO release and RHI improvements are statistically well-supported.

Areas for Improvement:

  • The results should include effect sizes in addition to p-values to provide a clearer understanding of the clinical relevance.
  • Some figures (e.g., Figures 2 and 3) could be optimized with clearer labeling and legend descriptions to enhance readability.
  1. Discussion

Strengths:

  • Well-integrated discussion with reference to existing literature.
  • Addresses the physiological mechanisms behind BPFG and L-citrulline synergy in NO bioavailability and endothelial function.
  • Highlights both strengths and limitations of the study.

Areas for Improvement:

  • The discussion on oxidative stress and NO degradation should be more detailed and linked to previous mechanistic studies.
  • The practical implications for athletes, including recommended dosages and potential side effects, should be elaborated.
  • Future research directions should be clearly outlined, particularly regarding the potential dose-response relationship and long-term effects.
  1. Conclusion

Strengths:

  • The conclusion effectively summarizes key findings and their significance.

Areas for Improvement:

  • The final statements should reinforce the translational application of the study findings to real-world athletic training and supplementation strategies.
  1. References

Strengths:

  • Citations are relevant and recent.

Areas for Improvement:

  • Ensure that all cited studies are appropriately formatted according to the journal’s referencing style.
  • A few more references supporting the biochemical pathways discussed would strengthen the argument.

Final Recommendation:

Minor revisions required

This study is well-conceived and methodologically sound, with results that contribute meaningfully to the field of sports nutrition. However, minor revisions are necessary to clarify methodological details, improve statistical reporting, and enhance discussion coherence. Once these points are addressed, the manuscript will be well-suited for publication in Nutrients.

Reviewer 2 Report

Comments and Suggestions for Authors

In this manuscript (ID# nutrients-3530691), entitled “Antioxidant properties and enhanced vasodilation contribute to the potentiation of L-citrulline effects in athletes mediated by combination with Bergamot Polyphenolic Fraction Gold”, authors Mollace et al studied the effect of Bergamot Polyphenolic Fraction Gold (BPFG) and L-citrulline on nitric oxide production, oxidative stress, endothelial function under rest and exercise conditions. Their results have demonstrated that BPFG and/or L-citrulline increased nitric oxide production; improved endothelial function, and attenuated exercise-induced oxidative stress. They conclude that BPFG + L-citrulline supplementation in athletes resulted in greater skeletal muscle blood flow and oxygen uptake, improving athletic performance. However, there are several major concerns, which are listed in the following paragraphs:

  1. Table 1: the variation of each group in age, body weight, echocardiograph should be analyzed to make sure no significant difference among those groups.
  2. The figure and table in the supplementary file should be statistical analyzed; and please provide title and detail legends.
  3. How did you define the athlete or they are just healthy individuals?  
  4. Effects of BPFG and L-citrulline on endothelial function and oxidative stress have been studied previously. Combination of BPFG and L-citrulline generated synergistic effect or just additive effect? What are the molecular mechanisms underlying this combination?
  5. The title is confusing, I suggest to update. Abstract should be written concisely. In the Introduction section, please clearly describe what is new (novelty) in this study.

Round 2

Reviewer 2 Report

Comments and Suggestions for Authors

The revised manuscript has been improved significantly. No further recommendation.